# Loneliness and Problematic Internet Use during COVID-19 Lock-Down

**DOI:** 10.3390/bs11010005

**Published:** 2021-01-06

**Authors:** Hasah Alheneidi, Loulwah AlSumait, Dalal AlSumait, Andrew P. Smith

**Affiliations:** 1The Social Development Office, Ministry of Social Affairs and Labor, Kuwait City 13150, Kuwait; h.alhunaidi@gmail.com; 2Department of Information Science, Kuwait University, Kuwait City 25944, Kuwait; Loulwah.alsumait@ku.edu.kw; 3Faculty of Education, Kuwait University, Kuwait City 25944, Kuwait; Dalal.alsumit@ku.edu.kw; 4School of Psychology, Cardiff University, Cardiff CF10 3AS, UK

**Keywords:** problematic internet use, internet addiction, loneliness, quarantine, lock-down, COVID-19, Middle East

## Abstract

(1) Background: During the COVID-19 pandemic, strict lock-down and quarantine were widely imposed by most governments to minimize the spread of the virus. Previous studies have investigated the consequences of the quarantine and social isolation on mental health and the present study examines loneliness and problematic internet use. (2) Methods: The current research used a cross-sectional survey during a lock-down phase of the COVID-19 pandemic. A sample of 593 participants from the Middle East region (Kuwait, Saudi Arabia) were tested using the short form of the Revised UCLA Loneliness Scale and the Internet Addiction Test. (3) Results: Results from regression analyses showed an association between loneliness and Problematic Internet Use (PIU), and an association between loneliness and the number of hours spent online. Younger participants reported greater loneliness. The quality of the relationship with the person(s) with whom they were spending their lock-down was also correlated with loneliness. Those who reported greater loneliness also obtained frequent news about the pandemic from social media. Problematic internet use was associated with loneliness and the predictors of loneliness. ANOVA analyses showed a dose-response between the predictors and PIU. (4) Conclusions: This study highlights the influence of the social characteristics of the local culture during the COVID-19 lock-down on feelings of loneliness and on PIU.

## 1. Introduction

The rapid spread of the Severe Acute Respiratory Syndrome Coronavirus 2 has prompted governments around the world to take drastic measures of lock-down and home/institutional quarantine to control the Coronavirus disease (COVID-19). These measures placed millions of people under daily stressful situations of social isolation and limited mobility, along with high levels of fear and uncertainty for extended periods. By the time this paper was written, the average days of 138 lock-downs in 80 countries/states was 48 days with a confirmed maximum duration of 83 days. As a result, calls to define the threats to mental health have been raised among psychologists and sociological researchers and service providers [1]. Considerable research [2,3,4,5,6,7,8,9,10,11,12,13,14,15,16] has focused on the mental health of people during the COVID-19 pandemic examining the effects of fear of infection, isolation and economic insecurity on their well-being (anxiety, depression, and stress).

Given the buffering role of Information and Communication Technology (ICT) on the influence of social isolation on mental health, and as an immediate mitigation measure, mental health advisors recommended increasing the use of the ICT to reduce the feelings of anxiety and social isolation [17,18,19]. Holmes, O’Connor, Perry et al. [19] reported the possible benefits of online communications during the pandemic. The systematic review in Brooks, Webster, Smith et al. [1] also found that online communications could reduce feelings of isolation, stress, and panic. Recent research [18] has shown that higher use of the internet was a predictor of higher levels of social support and decreased loneliness in older adults under social isolation due to COVID-19. Mucci, Mucci and Diolaiuti [17] emphasized the need for immediate online telemedicine and online social support interventions to aid people under the stressed quarantine conditions, yet warned of potential problematic outcomes due to excessive use. The term Problematic Internet Use (PIU) was first defined by Beard and Wolf [20] as excessive internet use that results in negative consequences on the users’ life in different aspects: social, psychological, academic, or any other life aspect [20]. Findings showed that internet addiction and PIU were associated with conditions such as loneliness, social withdrawal, emotional instability, depression, anxiety, low self-esteem, and other addictive behaviors [21,22,23,24].

Social isolation may also increase PIU, and research has shown that there has been an increase in problematic pornography use and online gaming during the COVID-19 crisis [25,26]. Research from China has also demonstrated that PIU increased during COVID-19 lock-down [27]. The association of PIU or internet addiction and loneliness has been demonstrated in previous studies, as has its association with low social support and poor family functioning [28,29,30,31,32]. Nevertheless, internet addicts who experience loneliness use the internet as a gateway [33,34,35], and Yao and Zhong [36] showed a repetitive cycle of loneliness and internet addiction.

### Purpose of the Study 

The first aim of the present study was to determine the risk factors for loneliness during the lock-down period of COVID-19. The second aim was to examine risk factors for PIU during the same period. These potential risk factors included features of the psychosocial context and also loneliness and its associated predictors. In addition to investigating these topics, the study focused on these issues in the Middle East where there is little literature both before and during the COVID-19 pandemic. Cheng and Li [37] conducted a cross-nation variation study in internet addiction based on published results of Young’s Questionnaire on internet addiction (IAT). The study included two nations from the Middle East and focused on the relationship of PIU scores to GDP per capita, life satisfaction, and specific national indices of environmental quality. In contrast, the work of Bener and Bhugra [38] studied the prevalence of PIU in high school and college students in the State of Qatar and its association to negative lifestyle and risk factors for depression, such as sleeping, eating disorders, and physical inactivity. Their study focused on adolescents and young adults, and the link between PIU and loneliness was not investigated. The present research had a more diverse sample of adults and also investigated quarantine-related factors.

The current study investigated risk factors for loneliness and PIU, and their association, using the short form of the Revised UCLA Loneliness Scale (ULS-6) with PIU scores based on the Internet Addiction Test (IAT). The study involved an online survey of a sample of 593 participants. Univariate analyses involved correlation and one-way ANOVA with loneliness and PIU scores as the dependent variables. Multi-variate analyses involved regression to identify the predictors of loneliness and PIU, and ANOVA analyses to examine dose-response. The first hypothesis tested was that loneliness would be associated with the age of the person, quality of the relationship with the person(s) sharing lock-down, and information obtained about the pandemic. The second hypothesis was that loneliness would be associated with hours on the internet and PIU.

## 2. Materials and Methods

The study was conducted online using the Qualtrics platform between the middle of April and first week of May 2020. It investigated the cross-sectional association between loneliness, PIU, and lock-down during COVID-19 using a multi-variate approach.

### 2.1. Measures

The survey consisted of questions displayed in English and Arabic designed to reach the largest number of participants. The questionnaire included the *Internet Addiction Test* IAT, [39] which consists of 20 items to examine the participant’s internet use during the last month for non-academic and non-job-related purposes, and measuring addiction based on the pathological gambling criteria of DSM-IV [35]. The participant answered the questions using Likert scales (0 = not applicable to 5 = always). The sum of scores determine three types of internet users, reflecting their internet dependency: controlled internet user, problematic internet user, and internet addicts. The Arabic version of the scale was used. The short form of the Revised UCLA Loneliness Scale (ULS-6) was used in this study. It is one of the most widely-used scales to measure loneliness and social isolation [40]. The original scale included 20 statements while the ULS-6 consists of a subset of six items [41]. The participants answered the items on a 4-point Likert scale ranging from (1 = never to 4 = often). Higher score indicated higher loneliness. The ULS-6 was translated to Arabic for this study, and a pilot study on 19 adults was conducted to test the validity. The reliability of the translated scale was examined with a Cronbach’s standardized alpha which was 0.76. The Arabic translations of the IAT and ULS-6 items are shown in the Appendix A.

Demographic data were collected on age, gender, nationality, education level, marital status, number of children, job sector, and job security. COVID-19-related questions included: whether the participant was infected by COVID-19, whether a family member was infected, and whether the participant or family members were working on the front-line during the COVID-19 pandemic. Lock-down-related questions asked about how committed the participant was to the lock-down rules, awareness of the importance of the lock-down and social distancing, number of people with whom the participant was spending the quarantine or lock-down, and a rating of the quality of the relationship with that person(s). Participants were also asked about their main internet activity, number of hours spent online, and if they carried out offline activities during the lock-down. Collecting demographic data was important in order to understand the role of the other factors and control their association and influence on loneliness and PIU during the COVID-19 lock-down period.

### 2.2. Participants

In defining the appropriate sample size, the Tabachnick and Fidell (2014) equation [42] was taken into consideration. Tabachnick and Fidell (2014) suggested the following formula for determining sample size, based on the effect size and the number of independent variables to be used in the regression analyses: N ≥ 50 + 8m (m = number of independent variables). A medium-size relationship between dependent and independent variable was assumed, with α = 0.05, β = 0.20 and 15 independent variables in the regression model, N ≥ 50 + (8) (15) = 170. A sample size of at least 170 would, therefore, be appropriate.

The study sample were Arabs aged 18 years and over. The inclusion criteria were internet users aged 18 years or over. The sample were recruited through social media (WhatsApp and Twitter), starting with family members and friends of the authors. In total, 857 responses were collected but some respondents were aged under 18 years, and others were only partially completed. A total of 593 valid questionnaires were collected. The participants participated in the study of their own volition, and were given a web-based consent form. 68% were female, 63.9% were aged between 18–35 years and 53.3% were single, 41.3% were married, and 5.1% were divorced. Questions about the participant’s educational level revealed 65.1% of the participants had bachelor degrees, 16.8% had masters and PhD degrees, 6.1% had diplomas, and nearly 12% had high school diplomas or below.

### 2.3. Analysis Strategy

SPSS 25.00 was used to conduct all statistical analyses. Data met the assumption of normality. The reliability of the scales was tested using Cronbach’s alpha coefficients. Pearson univariate correlations were conducted to assess the strength of the associations of the ULS-6 loneliness score, the IAT score and the continuous demographic and lock-down variables. Categorical variables were entered into one-way ANOVAS with loneliness and IAT scores as dependent variables. Multi-variate analyses involved separate regressions determining the predictors of loneliness and IAT scores. ANOVAS were then carried out with continuous variables split into tertiles. This was to determine whether there were significant interactions between variables, and to examine dose-response, which provides an indication of causal relationships.

## 3. Results

### 3.1. Reliability

Cronbach’s alpha coefficients were calculated for the IAT and the ULS-6 loneliness tests. Cronbach’s alpha coefficients were evaluated using the guidelines given by George and Mallery [43], which interpreted Cronbach’s alpha scores as: >0.9 excellent, >0.8 good, >0.7 acceptable, >0.6 questionable, >0.5 poor, and ≤0.5 unacceptable. The items for the UCLA loneliness scale-6 had a Cronbach’s alpha coefficient of 0.83, indicating good reliability. The items for the IAT had a Cronbach’s alpha coefficient of 0.90, indicating excellent reliability.

### 3.2. Frequency of Lock-Down-Related Ratings

73.2% of the sample were fully committed to the lock-down.99.3% understood the importance of the lock-down and social distancing.62.4% rated their relationship with those whom they were locked down with as excellent or good.52.8% were spending the lock-down with six or more people.5.6% were spending their lock-down in a room or a narrow apartment.17.4% were spending their lock-down in a medium to large apartment.10.6% were spending their lock-down in an apartment in their family house.58.7% were spending their lock-down in a house.7.8% were spending their lock-down in a farm, or in an urban area.

### 3.3. Frequency of Internet Use-Related Variables

13.7% spent two hours or less online daily.18% spent five hours online daily.21.7% spent six to seven hours online daily.31.5% spent eight hours or more online daily.31.9% worked or studied online regularly.53.6% used social networks as their main internet activity.26% had rarely or never carried out offline activities during the lock-down.

### 3.4. Correlations

Loneliness scores were significantly correlated with IAT scores (*r* = 0.43, *p* < 0.001). Loneliness was significantly correlated with the quality of the relation with whom the participant was spending the lock-down with (*r* = −0.35, *p* < 0.001). Loneliness was significantly correlated with numbers of hours spent online (*r* = 0.260, *p* < 0.001) and negatively correlated with age (*r* = −0.22, *p* < 0.001).

### 3.5. Categorical Variables

Loneliness and PIU were significantly associated with:Being single.Poor job security.Smaller accommodation.Getting news about COVID from social media.

These effects are shown in Table 1.

There were no significant effects of gender, education, job type, working or studying online, being on the frontline, being infected, having a family member who was infected, commitment to quarantine, perception of the importance of quarantine, and number of people/children sharing quarantine.

### 3.6. Regression Analyses

Two regression analyses were carried out. The predictor variables were those that were significant univariate predictors of loneliness and IAT scores. The first dependent variable was loneliness. The results of this regression are shown in Table 2.

The significant predictors were: being younger, having a poor relationship with the person/people sharing quarantine, spending long hours on the internet and getting news about COVID-19 from social media. The second regression had IAT scores as the dependent variable, and the results are shown in Table 3. The significant predictors were: loneliness, having a poor relationship with the person/people sharing lock-down, spending long hours on the internet, spending little time on other activities, and getting news about COVID-19 from social media.

### 3.7. ANOVAS

The regression analyses did not consider interactions between the predictor variables. This was examined in ANOVAS, one with loneliness as the dependent variable and the other with the IAT score as the dependent variable. None of the interactions were significant. 

In order to examine dose-response, the continuous variables were split into tertiles and these new variables were included in the ANOVAS. There were significant linear dose-response relationships for loneliness and the quality of the relationship with the person sharing lock-down and hours spent on the internet. This is shown in Table 4.

A similar pattern of dose-response was also found for these variables and IAT scores (see Table 4). Loneliness showed a clear dose-response relationship with the IAT scores, and this is shown in Figure 1.

## 4. Discussion and Conclusions

The present study showed the prevalence of lock-down-related variables, loneliness, and internet use. The results showed that 53.2% of the participants spent six hours or more a day on the internet, out of which 31.5% were online more than 8 h during lock-down. The results confirmed the hypothesis about predictors of loneliness. Demographic variables were related to loneliness, with being single, younger and living in small accommodation being associated with greater loneliness. However, the multivariate regressions showed that the main predictors of loneliness were the quality of the relationship with people sharing lock-down and hours on the internet. The results confirmed the hypothesis about associations between internet use and loneliness. The loneliness score was highly correlated with the number of hours spent online, which confirms the assumption of Caplan [44] that internet use may be high due to the feelings of loneliness and low social support. Yet, the result conflicts with the findings of Girdhar, Srivastava and Sethi [18] who found that high use of internet predicts low loneliness during lock-down.

The IAT scores were, not surprisingly, associated with hours on the internet, getting news from social media, and fewer offline interests. The correlation and regression results showed significant associations between loneliness and IAT scores. IAT scores were also significantly correlated with the quality of the relationship with whom the person was spending the lock-down with. This finding supports previous results showing that problematic internet users have low social skills and social support. Problematic internet users also scored higher in social media use which reflects the social need of communicating with others. However, the preference of problematic internet users for virtual relationships could be related to lack of social skills, low self-esteem, and isolation. This result confirms the findings of Young [39] that internet addicts prefer using the internet rather than spending time with their family members or spouses. In the present study, few would be classified as internet addicts, which suggests that earlier findings also apply to those with lower IAT scores.

A major limitation of the study was the use of a convenience sample which limits the extent to which the results generalize to the population as a whole. Another limitation of the present study was that it was cross-sectional, which makes it difficult to infer causality. One method of getting an indication of possible causal mechanisms is to examine dose-response. This was done here by splitting predictor variables into tertiles and examining the associations with loneliness and IAT scores. These analyses showed a linear dose-response and further longitudinal research is now required to consider causation in more detail. Interpretation of the present results is also rather difficult in that it is unclear whether it is the presence of COVID-19 or lock-down that affect loneliness and PIU. The present research also needs to be extended to examine the associations of the variables examined here with mental well-being. There is extensive literature on PIU and mental health, and also changes in mental health during COVID-19. These findings, and the results of the present study, make it plausible to suggest that loneliness and PIU may be important risk factors for mental health problems during the COVID-19 pandemic.

## Figures and Tables

**Figure 1 behavsci-11-00005-f001:**
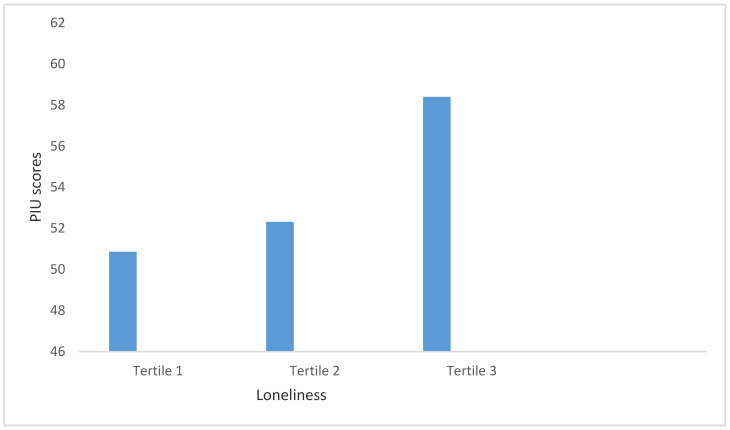
Dose-response relationship between loneliness and PIU.

**Table 1 behavsci-11-00005-t001:** Significant associations between categorical variables, loneliness and PIU (Problematic Internet Use).

	Loneliness (Mean, s.d., p)	PIU (Mean, s.d., p)
1. Marital status		
Single	35.7 (3.8) *p* < 0.001	52.2 (16.0) *p* < 0.001
Married	33.9 (3.7)	
2. Job security		
Good	34.5 (3.7) *p* < 0.005	48.0 (15.6) *p* < 0.005
Poor	35.8 (3.9)	52.7 (15.8)
3. Accommodation		
Small apartment	36.5 (3.8) *p* < 0.05	57.7 (15.7) *p* < 0.01
Medium apartment	35.0 (3.7)	47.8 (13.4)
House	34.5 (3.7)	49.4 (16.7)

**Table 2 behavsci-11-00005-t002:** Regression showing predictors of loneliness.

Coefficients					
Model	Unstandardized Coefficients		Standardized Coefficients	t	Sig.
	B	Std. Error	Beta		
(Constant)	37.881	1.670		22.681	0.000
Age	−0.348	0.154	−0.111	−2.256	0.024
Marital Status	−0.286	0.287	−0.047	−0.997	0.319
Job security	0.650	0.372	0.069	1.748	0.081
Good relationship with people in isolation	−0.047	0.008	−0.244	−6.062	0.000
Accommodation	−0.060	0.150	−0.016	−0.402	0.688
News source	0.807	0.331	0.096	2.437	0.015
Other interests	−0.064	0.175	−0.015	−0.363	0.716
Hours on internet	0.262	0.084	0.139	3.110	0.002

**Table 3 behavsci-11-00005-t003:** Regression showing predictors of problematic internet use (IAT (Internet Addiction Test) scores).

Model	Unstandardized Coefficients		Standardized Coefficients	t	Sig.
	B	Std. Error	Beta		
(Constant)	−26.486	8.060		−3.286	0.001
Age	−0.649	0.541	−0.050	−1.199	0.231
Marital Status	2.616	1.000	0.103	2.615	0.009
Job security	1.791	1.300	0.046	1.378	0.169
Good relationship with others in isolation	−0.064	0.028	−0.079	−2.265	0.024
Accommodation	0.856	0.524	0.056	1.634	0.103
News source	4.082	1.160	0.117	3.518	0.000
No other interests	3.398	0.612	0.192	5.552	0.000
Hours on internet	2.811	0.297	0.360	9.480	0.000
Loneliness	1.148	0.147	0.277	7.806	0.000

**Table 4 behavsci-11-00005-t004:** Dose-response relationships between quality of relationship, hours on internet, loneliness and PIU.

	Loneliness (Mean, s.e., p)	PIU (Mean, s.e., p)
1. Quality of relationship with people in isolation		
Tertile 1 (poor)	36.5 (0.48) *p* < 0.005	56.6 (1.78) *p* < 0.001
Tertile 2	35.0 (0.47)	54.5 (1.73)
Tertile 3 (good)	34.0 (0.47)	50.5 (1.74)
2. Hours on internet		
Tertile 1 (low)	34.5 (0.47) *p* < 0.005	46.3 (1.75) *p* < 0.001
Tertile 2	35.2 (0.45)	52.6 (1.66)
Tertile 3 (high)	35.8 (0.51)	62.7 (1.87)

## Data Availability

Requests for copies of the data should be sent to Professor A. P. Smith, smithap@cardiff.ac.uk.

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
