# Peer review of "Loneliness and Problematic Internet Use during COVID-19 Lock-Down"

_behavsci, 2021, doi:10.3390/bs11010005_

Round 1

Reviewer 1 Report

This work provides interest to understand the effect of problematic Internet use and loneliness during the closures and quarantines imposed during the COIVD-19 pandemic in the Middle East region (Kuwait and Saudi Arabia). To do this, the authors applied the UCLA Revised Loneliness Scale and the Internet Addiction Test on a sizeable sample of 593 participants. Their study findings show an association between loneliness and problematic Internet use. As well as an association between loneliness and the number of hours that participants were online. In general, I consider that the theme of the manuscript is adequate, interesting and current.
Also, the title of the manuscript and the abstract are adequate. So is the state of the art and the theoretical framework consulted by the authors.
In relation to the methodological section, I recommend that the authors explicitly state the hypotheses of their study in the manuscript. Likewise, the degree of compliance with them should be addressed in the conclusions section. Regarding statistical analyzes, I consider that they are adequate and well focused in relation to the objectives established in the investigation. However, the authors should provide more detailed information on the recruitment procedure for study participants.

Author Response

We thank the reviewer for the constructive comments.

The hypotheses of the study are now added (lines 87-90) and the extent to which they have been verified added to the conclusions (lines 253-254; 257-258).

More details of recruitment of the sample have also been added (lines 137-140).

Reviewer 2 Report

Thank you for allowing me to review this study about loneliness and internet use in the context of Covid19-related lockdown. Overall, the study seems to be well prepared and conducted. The introduction was sufficiently informative to state the problem.

My major concerns is that the study was based on a convenience sample, which limits generalizability of the findings. Also, the study – including the ‘dose-response’ examination - does not allow to draw any causal conclusions. It is not known if Covid19, or the related restrictions, have affected participants’ feelings of loneliness or internet use. This could be more clearly stated in the limitations.

Abstract: Please mention regression analysis in the methods of analysis.

Methods

It is not very clear who were the participants in the study. Please describe the inclusion and exclusion criteria and how and when participants were recruited.

What is known of participants who started the survey but did not complete it?

Good luck with the revision.

Author Response

We thank the review for the comments which have led to an improved paper.

The problem of using a convenience sample is added to the limitations (lines 274-275).

The problem of effects being due to COVID-19 or lockdown is also added (lines 280-282). 

Regression analyses are now mentioned in the abstract (line 18).

More details of the recruitment of the sample are now given (lines 137-140).